# Projecting contact matrices in 177 geographical regions: An update and comparison with empirical data for the COVID-19 era

**Kiesha Prem**[1,2], **Kevin van Zandvoort**[1], **Petra Klepac**[1], **Rosalind M. Eggo**[1], **Nicholas G. Davies**[1], **Centre for the Mathematical Modelling of Infectious Diseases COVID-19 Working Group**[¶], **Alex R. Cook**[2], **Mark Jit**[1] *

**1** Centre for Mathematical Modelling of Infectious Diseases, London School of Hygiene & Tropical Medicine, London, United Kingdom, **2** Saw Swee Hock School of Public Health, National University of Singapore, Singapore

¶ Membership of the Centre for the Mathematical Modelling of Infectious Diseases COVID-19 Working Group is listed in the Acknowledgments.
* mark.jit@lshtm.ac.uk

**Data Availability Statement:** All data used in this study can be downloaded from the cited

## Abstract

Mathematical models have played a key role in understanding the spread of directly-transmissible infectious diseases such as Coronavirus Disease 2019 (COVID-19), as well as the effectiveness of public health responses. As the risk of contracting directly-transmitted infections depends on who interacts with whom, mathematical models often use contact matrices to characterise the spread of infectious pathogens. These contact matrices are usually generated from diary-based contact surveys. However, the majority of places in the world do not have representative empirical contact studies, so synthetic contact matrices have been constructed using more widely available setting-specific survey data on household, school, classroom, and workplace composition combined with empirical data on contact patterns in Europe. In 2017, the largest set of synthetic contact matrices to date were published for 152 geographical locations. In this study, we update these matrices for populations up to age 80 with the most recent data and extend our analysis to 177 geographical locations. Due to the observed geographic differences within countries, we also quantify contact patterns in rural and urban settings where data is available. Further, we compare both the 2017 and 2020 synthetic matrices to out-of-sample empirically-constructed contact matrices, and explore the effects of using both the empirical and synthetic contact matrices when modelling physical distancing interventions for the COVID-19 pandemic. We found that the synthetic contact matrices show qualitative similarities to the contact patterns in the empirically-constructed contact matrices. Models parameterised with the empirical and synthetic matrices generated similar findings with few differences observed in age groups where the empirical matrices have missing or aggregated age groups. This finding means that synthetic contact matrices may be used in modelling outbreaks in settings for which empirical studies have yet to be conducted.

references. The codes used to generate these analyses and the updated synthetic matrices are available at https://github.com/kieshaprem/synthetic-contact-matrices DOI: 10.5281/zenodo.4889500.

**Funding:** The following funding sources are acknowledged as providing funding for the named authors. KP, PK and MJ were partly funded by the Bill & Melinda Gates Foundation (INV-003174). MJ and NGD were partly funded by the National Institute for Health Research (NIHR; NIHR200929) MJ was partly funded by NIHR using UK aid from the UK Government to support global health research (16/137/109). The views expressed in this publication are those of the author(s) and not necessarily those of the NIHR or the UK Department of Health and Social Care. KvZ was partly funded by DfID/Wellcome Trust (Epidemic Preparedness Coronavirus research programme 221303/Z/20/Z) and DfID/Wellcome Trust/NIHR (Elrha R2HC/UK DFID/Wellcome Trust/NIHR). KP, PK and MJ were partly funded by the European Union's Horizon 2020 research and innovation programme - project EpiPose (101003688). RME was partly funded by HDR UK (MR/S003975/1) and UK MRC (MC_PC 19065). PK was partly funded by the Royal Society (RP\EA\180004). ARC was partly funded by the Singapore's National Medical Research Council (COVID19RF-004, NMRC/CG/C026/2017_NUHS). The following funding sources are acknowledged as providing funding for the working group authors. Alan Turing Institute (AE). BBSRC LIDP (BB/M009513/1: DS). This research was partly funded by the Bill & Melinda Gates Foundation (INV-001754: MQ; INV-003174: YL; NTD Modelling Consortium OPP1184344: CABP, GM; OPP1180644: SRP; OPP1183986: ESN; OPP1191821: KO'R, MA). DFID/Wellcome Trust (Epidemic Preparedness Coronavirus research programme 221303/Z/20/Z: CABP). DTRA (HDTRA1-18-1-0051: JWR). ERC Starting Grant (#757688: CJVA, KEA; #757699: JCE, RMGJH; 757699: MQ). This project has received funding from the European Union's Horizon 2020 research and innovation programme - project EpiPose (101003688: WJE, YL). This research was partly funded by the Global Challenges Research Fund (GCRF) project 'RECAP' managed through RCUK and ESRC (ES/P010873/1: AG, CIJ, TJ). Nakajima Foundation (AE). This research was partly funded by the National Institute for Health Research (NIHR) using UK aid from the UK Government to support global health research. The views expressed in this publication are those of the author(s) and not necessarily those of the NIHR or the UK Department of Health and Social Care (16/137/109: CD, FYS, YL; 16/137/109 & 16/

## Author summary

The risk of contracting a directly transmitted infectious disease such as the Coronavirus Disease 2019 (COVID-19) depends on who interacts with whom. Such person-to-person interactions vary by age and locations—e.g., at home, at work, at school, or in the community—due to the different social structures. These social structures, and thus contact patterns, vary across and within countries. Although social contact patterns can be measured using contact surveys, the majority of countries around the world, particularly low- and middle-income countries, lack nationally representative contact surveys. A simple way to present contact data is to use matrices where the elements represent the rate of contact between subgroups such as age groups represented by the columns and rows. In 2017, we generated age- and location-specific synthetic contact matrices for 152 geographical regions by adapting contact pattern data from eight European countries using country-specific data on household size, school and workplace composition. We have now updated these matrices with the most recent data (Demographic Household Surveys, World Bank, UN Population Division) extending the coverage to 177 geographical locations, covering 97.2% of the world's population. We also quantified contact patterns in rural and urban settings. When compared to out-of-sample empirically-measured contact patterns, we found that the synthetic matrices reproduce the main features of these contact patterns.

## Introduction

The emergence of the Severe Acute Respiratory Syndrome Coronavirus 2 (SARS-CoV-2) responsible for causing Coronavirus Disease 2019 (COVID-19) has affected the lives of billions worldwide [1]. SARS-CoV-2 is predominantly transmitted between people via respiratory droplets and, as such, the transmission dynamics are strongly influenced by the number and type of close contacts between infectious and susceptible individuals [2–7].

Mathematical models have played a key role in understanding both the spread of directly-transmissible infectious diseases such as COVID-19 [8–10] and the effectiveness of public health responses [11–16]. Since transmission events can rarely be directly observed and measured, most transmission models are based on the *social contact hypothesis* [17] which implies the risk of transmission between a susceptible and an infected individual be proportional to the rate of contact between them [18]. Rates of contact are known to differ according to characteristics such as the age, of both individuals, and the setting in which the contact takes place, such as the home, school or workplace; they are also commonly assortative, and infection may be concentrated in demographic segments as a result [17,19,20].

Age-structured models often define the rate of mixing between age groups through a *mixing matrix* where the elements represent the frequency of contact between two individuals from subgroups (such as age groups) represented by the columns and rows. Mixing matrices can be generated from surveys that record the number and type of contacts between people, such as the respondent-completed diaries used in the landmark POLYMOD contact pattern study, which measured social contact patterns in eight European countries [20]. However, the majority of countries around the world lack data from contact surveys that can be used to inform the mixing matrix. This problem is particularly acute in low- and lower-middle-income countries (LMICs), where only 4 studies are available, compared to 54 in high-income countries [21]. Our previous work [22] used country-specific data on household size, school, and workplace composition plus empirical contact data from the POLYMOD survey to generate age- and

136/46: BJQ; Health Protection Research Unit for Modelling Methodology HPRU-2012-10096: TJ; PR-OD-1017-20002: AR). Royal Society (Dorothy Hodgkin Fellowship: RL). UK DHSC/UK Aid/NIHR (ITCRZ 03010: HPG). UK MRC (LID DTP MR/N013638/1: GRGL, QJL; MR/P014658/1: GMK). Authors of this research receive funding from UK Public Health Rapid Support Team funded by the United Kingdom Department of Health and Social Care (TJ). Wellcome Trust (206250/Z/17/Z: AJK, TWR; 206471/Z/17/Z: OJB; 208812/Z/17/Z: SC, SFlasche; 210758/Z/18/Z: JDM, JH, NIB, SA, SFunk, SRM). The funders had no role in study design, data collection and analysis, decision to publish, or preparation of the manuscript.

**Competing interests:** The authors have declared that no competing interests exist.

location-specific contact matrices (synthetic contact matrices) to use in settings where social contact patterns had not yet been directly measured.

These synthetic contact matrices have been widely used in models of SARS-CoV-2 spread and the impact of interventions such as physical distancing which alter the pattern of contacts (e.g. [13]). Following the publication of our previous work, new empirical contact surveys have been conducted in LMICs (reviewed in [21]), full demographic data are now available for more countries for older age groups, which is particularly salient given the age-gradient in the severity of COVID-19 [23,24], and more recent household composition data are now available for more countries than before. Updating the matrices is particularly important since public health interventions during the pandemic, such as shielding, are often age-structured [25].

Geographic differences within countries have also been observed, with large early outbreaks in urban population centres such as Wuhan, New York, London and Madrid [26,27] spreading into more rural areas, which in many countries may lack the healthcare infrastructure to handle surges in severe cases. Tailored public health response in rural and urban settings may thus be called for to minimise unnecessary economic and social impacts. Assessing such policies requires differences between contact patterns in rural and urban environments to be quantified, which has previously been done only for a few countries [28–30]. In these studies, individuals in rural settings documented more contacts at home than their urban counterparts [28,29]. However, individuals in rural settings in Zimbabwe [30] reported a lower total number of contacts than those in peri-urban settings. The study in Southern China observed no qualitative difference in overall contact patterns between rural and urban populations [28].

In this paper, we update the synthetic contact matrices with the most recent data, comparing them to measured contact matrices, and develop customised contact matrices for rural and urban settings. We use these to explore the effects of physical distancing interventions for the COVID-19 pandemic in a transmission model.

## Materials and methods

### Updating country-specific demography and setting parameters

As in Prem et al. [22], we employed a Bayesian hierarchical modelling framework to estimate the age- and location-specific contact rates in each of the POLYMOD countries (Belgium, Germany, Finland, United Kingdom, Italy, Luxembourg, the Netherlands, Poland), accounting for repeat measurements of contacts made in different locations by the same individual. We model the number of contacts documented by individual $i$ at a particular location $L$ with an individual in age group $\alpha$, as $X_{i,\alpha}^{L} \sim \text{Po}(\mu_{a_{i},\alpha}^{L})$ where the mean parameter varies for each individual $i$, by $i$'s age, $a_i$, and by location, i.e: $\mu_{a_{i},\alpha}^{L} = \sigma_{i}\lambda_{a_{i},\alpha}^{L}$. The $\sigma_i$ parameter characterises differences in social activity levels between individuals i.e., the random effect belonging to individual $i$. The $\lambda_{a_{i},\alpha}^{L}$ parameter denotes the frequency of contact between individuals (or contact rate per day) from two age groups, $a$ and $\alpha$, at location $L$ and it is the key estimand. Because the number of contacts should be comparable for individuals of similar ages, we imposed smoothness between successive age groups for the $\lambda_{a_{i},\alpha}^{L}$ parameter as described in section A.8 in **S1 Text**. Noninformative prior distributions were assumed for all parameters in the model, as detailed in [22].

We updated the synthetic contact matrices [22] with more recent data on population age structure, household age structure of 43 countries with recent Demographic Household Surveys (DHS) [31] and socio-demographic factors for 177 geographical regions, including countries and some subnational regions such as the Hong Kong and Macau Special Administrative Regions (SARs) of the People's Republic of China. We include 14 country characteristics from the World Bank and United Nations Educational, Scientific and Cultural Organization

Institute for Statistics (UIS) databases: gross domestic product per capita, total fertility rate and adolescent fertility rate, population density, population growth rate, internet penetration rate, secondary school education attainment levels, as proxies of development, and under-five mortality rate, the life expectancy of males and females, mortality rates of males, risk of maternal death, mortality from road traffic injury, and the incidence of tuberculosis, as proxies for overall health in the country. The DHS provides nationally-representative household surveys with the largest dataset, from India, containing information on ~ 3 million individuals from about 600 000 households (see Table A in **S1 Text**). To project the household age structure for a geographical location with no available household data, we use a weighted mean of the population-adjusted household age structures of the POLYMOD and DHS countries as described in the **S1 Text**. Because the household age structures vary across countries in different stages of development and with different demographics, we use the updated 14 indicators, all standardized by z-scoring, to quantify the similarities between countries with and without household data to derive these weights. We internally validated the household age matrices using leave-one-out validation to verify these matrices describing household structure could be reverse-engineered for countries (POLYMOD and DHS) for which empirical household age matrices were available, as described in Prem et al. [22] and in **S1 Text**.

By accounting for the demographic structure, household structure (where known), and a variety of metrics including workforce participation and school enrolment, we then estimated contact patterns at home, work, school and other locations for non-POLYMOD countries. Specifically, the population age compositions for 177 geographical regions were obtained from the United Nations Population Division [32]. To derive the working population matrices for each geographical location $c$, we use the labour force participation rate by sex and 5-year age groups, $w_a^c$, for the 177 geographical regions from the International Labour Organization (ILO) [33].

When constructing the school-going population matrices, we use the country-specific pupil-to-teacher ratio in schools at various level of education (i.e., pre-primary, primary, secondary and tertiary), enrolment rates of students at various level of education, starting ages and number of years of schooling at various level of education from UIS [34] and the distribution of teachers by age from the Organisation for Economic Co-operation and Development (OECD) [35]. Using the country-specific data, we first estimate the number of students in each age group by education level. Together with the country-specific pupil-to-teacher ratio at each education level, distribution of teachers and workforce by age, we then project the number of teachers in each age group. Both students and teachers form the school-going population. The steps to construct both the working and school-going populations are detailed in **S1 Text**.

After projecting populations at home, work and school for the 177 geographical regions, we infer the synthetic age- and location-specific contact matrices (**S1 Text**). For contacts in other locations (not home, work or school), we adjusted the POLYMOD contact matrices with the country-specific population. We also compare the proportion of contacts at other locations measured from the empirical contact studies.

## Stratifying contacts by rural and urban areas

We stratified the age- and location-specific contact matrices according to rural and urban areas by the rural and urban population age compositions for all geographical regions of the world from the United Nations Population Division [36] (see [37] for urban and rural classification). The nationally-representative DHS household surveys additionally provide data for rural and urban areas [31], allowing us to derive rural-urban household age matrices. We compare the population age compositions and household age matrices in rural and urban settings of countries with stratified household data (**S1 Text** and **S2 Text**).

We assessed the age-specific labour force participation rates by rural and urban regions from ILO [38]. Using the differences in rural and urban schools' pupil-to-teacher ratio from OECD [39], we construct rural and urban school population matrices. These differences were available for 36 countries, and we assumed the OECD average for the regions without data. We also compare the mean total number of contacts among children (0–9-year-olds) and older adults (60–69-year-olds), as well as the basic reproduction number in rural and urban settings.

## Comparing synthetic matrices to empirical contact matrices

We extracted data from all contact surveys listed in the Zenodo social contact database [40] and directly from the published studies [41], and used them to construct empirical contact matrices using the socialmixr R package [42]. Data were available for 11 geographical locations: Shanghai and Hong Kong SAR, China [43,44], France [45], Kenya [46], Peru [47], the Russian Federation [48], South Africa [29,49], Vietnam [50], Zambia [29] and Zimbabwe [30]. We then compared each element of the empirical matrices with our synthetic matrices. We also compared the proportion of contacts in "Other" locations, since this was the only setting not directly informed by local data (other than population age structure) in the synthetic matrices. To understand potential sources of differences between the empirical and synthetic matrices as well as between empirical matrices between different regions, we extracted details of how each survey was conducted from the original publications.

Table 1 summarises the changes between the construction of the 2017 and 2020 synthetic matrices. Analyses were done in R version 3.6.2 [51], and the codes are deposited in https://doi.org/10.5281/zenodo.8383778.

**Table 1. Summary of the changes between the 2017 and 2020 synthetic matrices.**

| | 2020 synthetic matrices | 2017 synthetic matrices |
|---|---|---|
| Overall coverage | 177 geographical locations covering 97.2% of the world's population, including rural and urban settings | 152 geographical locations covering 95.9% of the world's population |
| Population age composition data | 2020 UN Population Division population demographic data for 177 geographical regions | 2015 UN Population Division population demographic data for 152 geographical regions |
| Urban and rural population age composition data | Stratified population demographic data by urban and rural settings for 177 geographical regions | Not considered |
| Country-specific household data | 51 countries: 34 additional low- and lower-middle-income countries | 17 countries |
| Country-specific urban and rural household data | Stratified household data by urban and rural settings (for 43 DHS countries) | Not considered |
| Country-specific labour force participation rate data by age | 177 geographical locations and stratified by urban and rural settings | 152 geographical locations |
| Country-specific school data | School data were curated at various levels of education: pre-primary, primary, secondary and tertiary. The updated school data included enrolment rates, average starting ages of schooling, number of years of schooling, pupil-to-teacher ratio at each education level from UNESCO Institute for Statistics for 177 geographical regions. | School data were curated at various levels of education: pre-primary, primary, secondary and tertiary. The country-specific school data included enrolment rates, pupil-to-teacher ratio at each education level from UNESCO Institute for Statistics for 152 geographical regions. |
| Country-specific urban and rural school data | Rural and urban differences in schools' pupil-to-teacher ratio for 36 countries from OECD | Not considered |
| Comparison between the synthetic matrices and out-of-sample empirically-constructed contact matrices | Comparison of mean contacts by age, full contact matrices, the proportion of contacts at other locations, and reduction in cases and age-specific infection attack rate using an age-stratified compartmental model of COVID-19. Social contact data from contact surveys were extracted from the Zenodo social contact database and directly from the published studies for 10 geographical locations. | Comparison of mean contacts by age. Contact matrices were assessed directly from the published studies in 5 geographical locations: Kenya (digitised image), Peru (digitised image), Russia, South Africa, and Vietnam. |

### Impact on modelling of interventions

We compare the difference in reduction of COVID-19 cases between using the empirical and synthetic matrices in models of COVID-19 epidemics in ten geographical regions—China, France, Hong Kong SAR, Kenya, Peru, the Russian Federation, South Africa, Uganda, Vietnam and Zimbabwe—using an age-stratified compartmental model [13,25]. We model an unmitigated epidemic and three intervention scenarios: 20% physical distancing, 50% physical distancing, and national lockdown. In all intervention scenarios, we assume a 50% reduction in transmission from individuals with clinical symptoms through self-isolation. In addition, we assume the following: (i) 20% physical distancing: 20% reduction in transmission outside of the household, (ii) 50% physical distancing: 50% reduction in transmission outside of the household, (iii) national lockdown: where we applied the pooled mean reduction in setting-specific contacts (i.e. at home, school, work, and other places) as observed in lockdowns implemented in several countries during the COVID-19 pandemic [53–57]. We considered six contact matrices when modelling the interventions to the COVID-19 pandemic: the empirically-constructed contact matrices at the study-year and adjusted for the 2020 population, the 2017 synthetic matrices, and the updated synthetic matrices at the national, rural, or urban settings. Synthetic matrices were implemented by assuming that the contacts made by the 75–80 years old age group were representative for those made by the 75+ years old age group in the model. More details of the model can be found in sections A.7 of **S1 Text**.

## Results

Twenty-five geographical regions were added to this study compared to the 2017 study. We also updated the population demographic data used for all countries including Namibia, Syrian Arab Republic, Republic of South Sudan, Kuwait, and Vanuatu where the proportion of individuals aged > 70 years was previously not recorded.

There were varied methods adopted in 11 contact surveys conducted to generate the empirical contact matrices covering 11 geographical locations (**Table 2**). The surveys differed substantially from each other and the original POLYMOD survey in sampling frames and survey methodology. Dodd et al. [29] measured social contacts among adults in South Africa and Zambia. Three surveys were conducted in exclusively rural regions [41,47,49], (including one in a remote highlands region [47]), three other surveys were conducted only in urban regions [43,44,48], and the remaining five surveys were conducted in a variety of urban and rural settings [29,30,45,46,50]. Although most studies adopted random or stratified sampling to recruit their respondents, a handful included convenience [43,47,48] and quota [44,45] sampling methods in their recruitment. In most contact diary approaches, contacts are categorised as physical contacts (e.g., skin-to-skin contacts) and nonphysical contacts (e.g. two-way conversations with three or more words in the physical presence of another person) [20]. They were equally split between studies that asked respondents to fill in surveys retrospectively [29,41,44,47,50] and prospectively [30,43,45,46,48,49].

The estimated proportions of contacts in other locations from POLYMOD contact survey largely match analogous figures in empirical contact studies from five geographical locations which report this—Shanghai and Hong Kong SAR, China; the Russian Federation; Peru; and Zimbabwe—but are higher than those from France for most ages (**Fig 1**). It is slightly higher in the synthetic matrices in adults (i.e., 20–40-year-olds) in Shanghai, Hong Kong and the Russian Federation, and slightly lower in older individuals (i.e., >60-year-olds) in Peru, but all other ages match closely.

The pronounced diagonals observed in all contact matrices are matched in the synthetic matrices (**Figs 2 and 3, S3 Text**), as are the secondary diagonals indicating the occurrence of

**Table 2. Description of empirical contact survey studies used to construct contact matrices.**

| Region, Country | Authors (year) | Study design | Study population | Sampling methods | Data collection method (mode) | Rural / Urban | Casual contacts (i.e., short-term) |
|---|---|---|---|---|---|---|---|
| Shanghai, China | Zhang et al. (2019) [43] | Prospective | General population (n = 965) | Convenience sample | Paper-diary (self-report and interview-based) | Urban | Participants were allowed to include group contacts |
| France | Béraud et al. (2015) [45] | Prospective | General population (n = 2033) | Quota sampling | Paper-diary (self-report) | Rural and urban | Participants were allowed to provide open-ended notes |
| Hong Kong SAR, China | Leung et al. (2017) [44] | Retrospective | General population (n = 1149) | Quota sampling | Paper-diary and online survey (self-report) | Urban | Participants were allowed to include group contacts |
| Kilifi, Kenya | Kiti et al. (2014) [46] | Prospective | General population (n = 568) | Stratified random sample | Paper-diary (self-report) | Rural and semi-urban | Undocumented |
| Highlands San Marcos, Cajamarca-Peru | Grijalva et al. (2015) [47] | Retrospective | General population (n = 588) | Convenience sample | Paper-diary (interview-based) | Rural | Participants were allowed to report unregistered contacts |
| Tomsk, Russia | Ajelli et al. (2017) [48] | Prospective | General population (n = 559) | Random and convenience samples | Paper-diary (self-report) | Urban | Participants were allowed to report unregistered contacts. |
| South Africa | Johnstone et al. (2011) [49] | Prospective | General population (n = 571) | Random sample | Paper-diary (self-report) | Rural | Undocumented |
| Sheema North Sub-District, Uganda | le Polain de Waroux et al. (2018) [41] | Retrospective | General population (n = 566) | Random sample | Paper-diary (interview-based) | Rural | Participants were allowed to report unregistered contacts |
| Red River Delta, North Vietnam | Horby et al. (2011) [50] | Retrospective | General population (n = 865) | Random sample | Paper-diary (interview-based) | Semi-rural | Participants were allowed to report unregistered contacts. |
| Zambia and South Africa | Dodd et al. (2016) [29] | Retrospective | Adults (>18 years) (n = 3582) | Random sample | Paper-diary (interview-based) | Rural and urban | Participants were asked by the interviewer |
| Manicaland, Zimbabwe | Melegaro et al. (2017) [30] | Prospective | General population (n = 2490) | Stratified random sample | Paper-diary (self-report) | Rural and peri-urban | Participants were asked by the interviewer |

intergenerational mixing. The updated synthetic contact matrices show close similarities to empirical matrices (median correlation between normalised synthetic and empirical matrices 0.84, interquartile range 0.76–0.88). In most geographical regions, both matrices are similar in terms of symmetry. However, there are a few places such as Zimbabwe and China (Shanghai) where the synthetic matrix is more symmetrical than the empirical matrix, as the latter shows more weight above the diagonal (young people report more contacts with old people than vice versa). The degree of symmetry of both synthetic and empirical matrices in each region is compared in Table E in S4 Text.

We reconstructed the empirical household age structures for the POLYMOD and DHS countries with high fidelity (median correlation between the observed and modelled household age matrix (HAM) 0.96, with an interquartile range 0.94–0.97) (See S1 Text section A.2.2 and S2 Text section B.1 for details sections A.2.2 and B.1 for details). The differences in the population and households age composition by rural and urban settings are presented in section B.2 in S2 Text.

In most countries, urban settings had a larger mean number of contacts among children (i.e., 0–9-years-old) (Fig 4). However, little difference was observed in the mean number of contacts among older adults (i.e. 60-69 years old) in rural and urban settings. The basic reproduction number in rural and urban settings are positively correlated (highest in high-income countries: r = 0.90, 95% confidence interval: 0.82–0.94) and higher in urban settings.

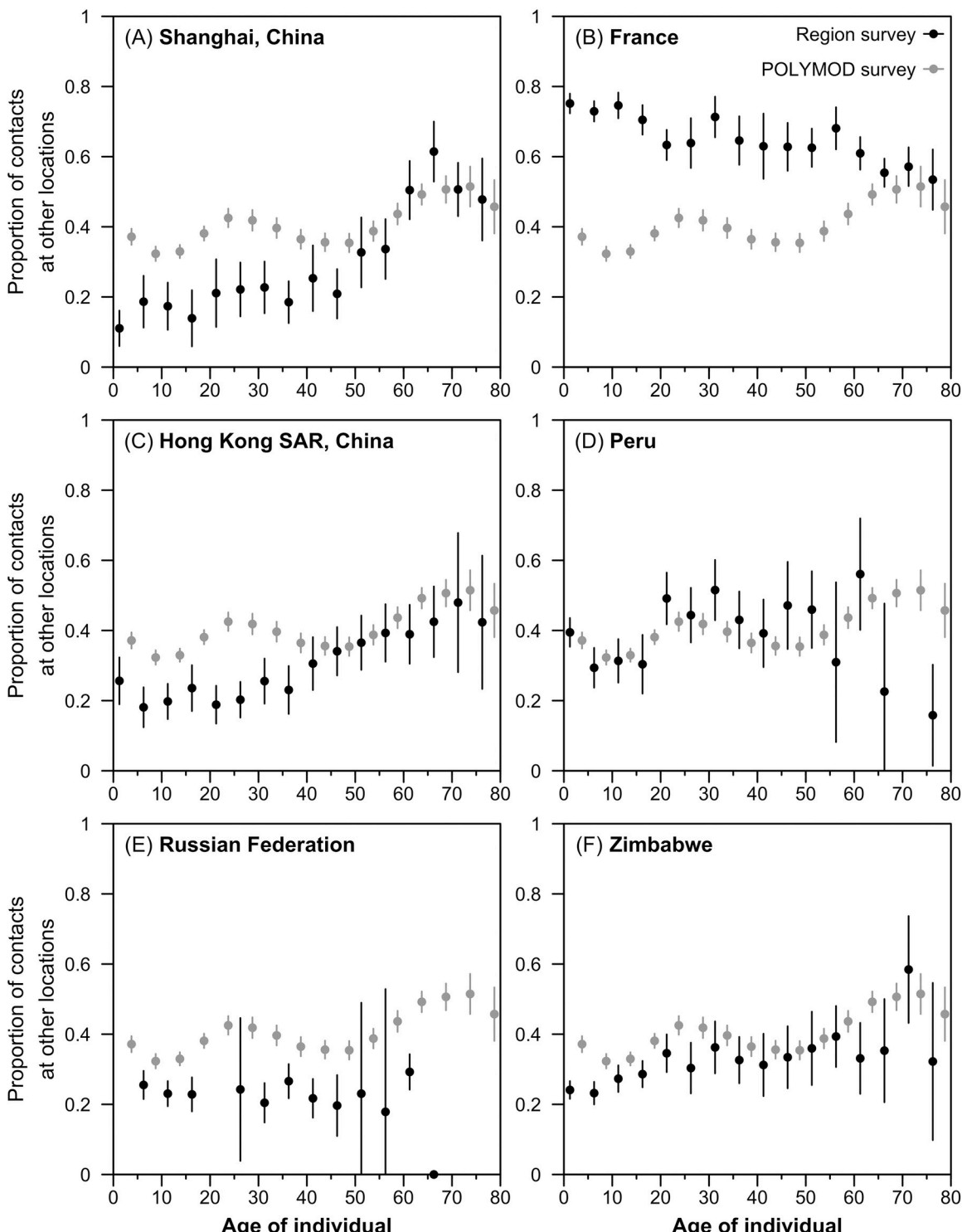

**Fig 1. Comparison of the estimated proportion of contacts at other locations for the empirical contact studies from six geographical regions and POLYMOD survey.** The estimated age-specific proportion of all contacts at other locations—transport, leisure, other locations—matrices from contact surveys at the country or geographical region (in black) are compared against that observed in the POLYMOD countries (in grey).

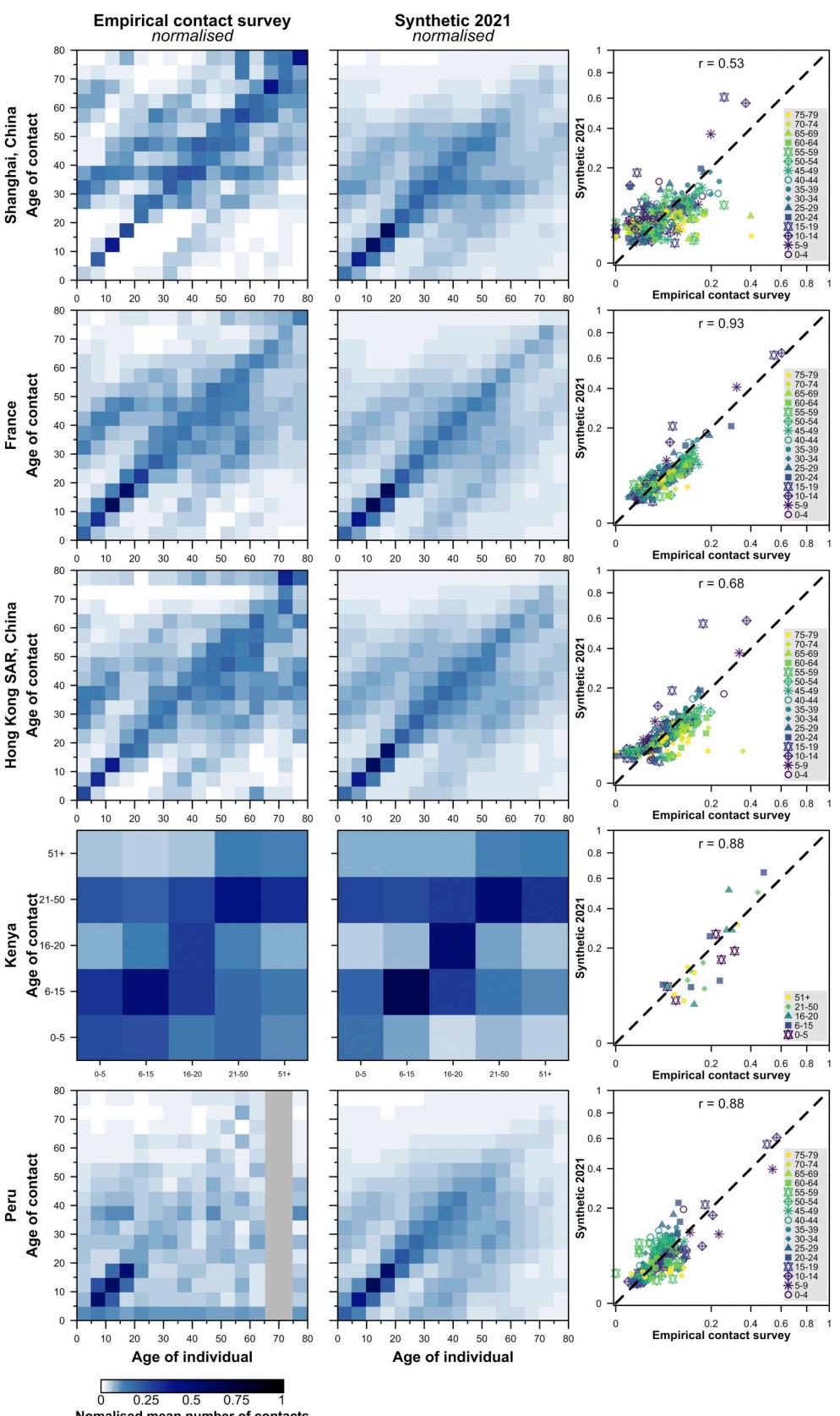

**Fig 2. Comparison of the normalised empirical and synthetic age-specific contact matrices in five geographical regions.** The empirical matrices collected from contact surveys, modelled synthetic contact matrices, and the scatter plots of the entries in the observed (x-axis) and modelled (y-axis) contact matrices are presented. The correlation between the empirical and synthetic matrices are shown. The matrices are normalised such that its dominant eigenvalue is 1. To match the population surveyed in the empirical studies, the contact matrices from rural settings of Kenya and Peru are presented; and the contact matrix from urban settings of China is presented. No data are available in the grey regions.

The choice of using synthetic or empirical matrices did not make a large difference to the infection attack rate for an unmitigated epidemic (Fig D in **S4 Text**), or to the overall number of severe COVID-19 cases predicted in a mathematical model of SARS-CoV-2 transmission and disease across the three physical distancing interventions (**Fig** 5 and Fig E in **S4 Text**). Where there were discrepancies, the relative magnitude of this discrepancy differed between countries. Differences were more marked in specific age groups (e.g. older people in Hong Kong SAR, Peru, Uganda, Vietnam and Zimbabwe; 10–20 year olds in China; 20–24 year olds in Russia). The largest age-related differences could potentially be attributed to particular features of empirical survey design such as missing (Peru, Russia) or aggregated (South Africa, Uganda, Vietnam) age groups, mode of questionnaire chosen by participants (Hong Kong SAR) and survey administration during school holidays (Zimbabwe) (See Table D in **S4 Text** for details).

## Discussion

Social mixing patterns have not been directly measured in most countries or regions within countries, particularly in low- and lower-middle-income settings. Synthetic contact matrices provide alternative age- and location-specific social mixing patterns for countries in different stages of sociodemographic and economic development [22]. The synthetic contact matrices presented here were derived by the amalgamation of several data sources and methods: (i) integration into a Bayesian hierarchical framework of age- and location-specific contact rates from eight European countries from the POLYMOD contact study; (ii) construction of age-structured populations at home, work, and school in many non-POLYMOD countries by combining household age-structure data from the POLYMOD study and DHS (which include mostly data from lower-income countries), socio-demographic factors from the UN Population Division and various international indicators; and (iii) projection of age-structured populations at home, work, and school and age- and location-specific contact matrices to other non-POLYMOD and non-DHS countries. Both empirical and synthetic contact matrices capture age-assortativity in mixing patterns; the pronounced primary diagonal highlights that individuals interact with others of similar age. Both also show secondary diagonals, approximately one generation apart, indicating parent-child interactions.

This paper provides a substantial update and improvement to previous synthetic matrices published in 2017 (**Table 1**). Improvements in the availability of demographic data globally have enabled us to provide validated approximations to age- and location-specific contact rates for 177 geographical regions covering 97.2% of the world's population, compared to 152 geographical regions covering 95.9% previously. Household data from 34 additional LMICs were included in the revision. We have also used the most recent data to build the working and school-going populations. We have extended the method to project contact patterns in rural and urban settings using country-specific urban and rural data. We find a higher positive correlation in mean contact rates and basic reproduction number in rural and urban settings of high-income countries, owing to the smaller rural-urban differences in these countries. Moreover, when assessing the consistency of results under different mixing assumptions

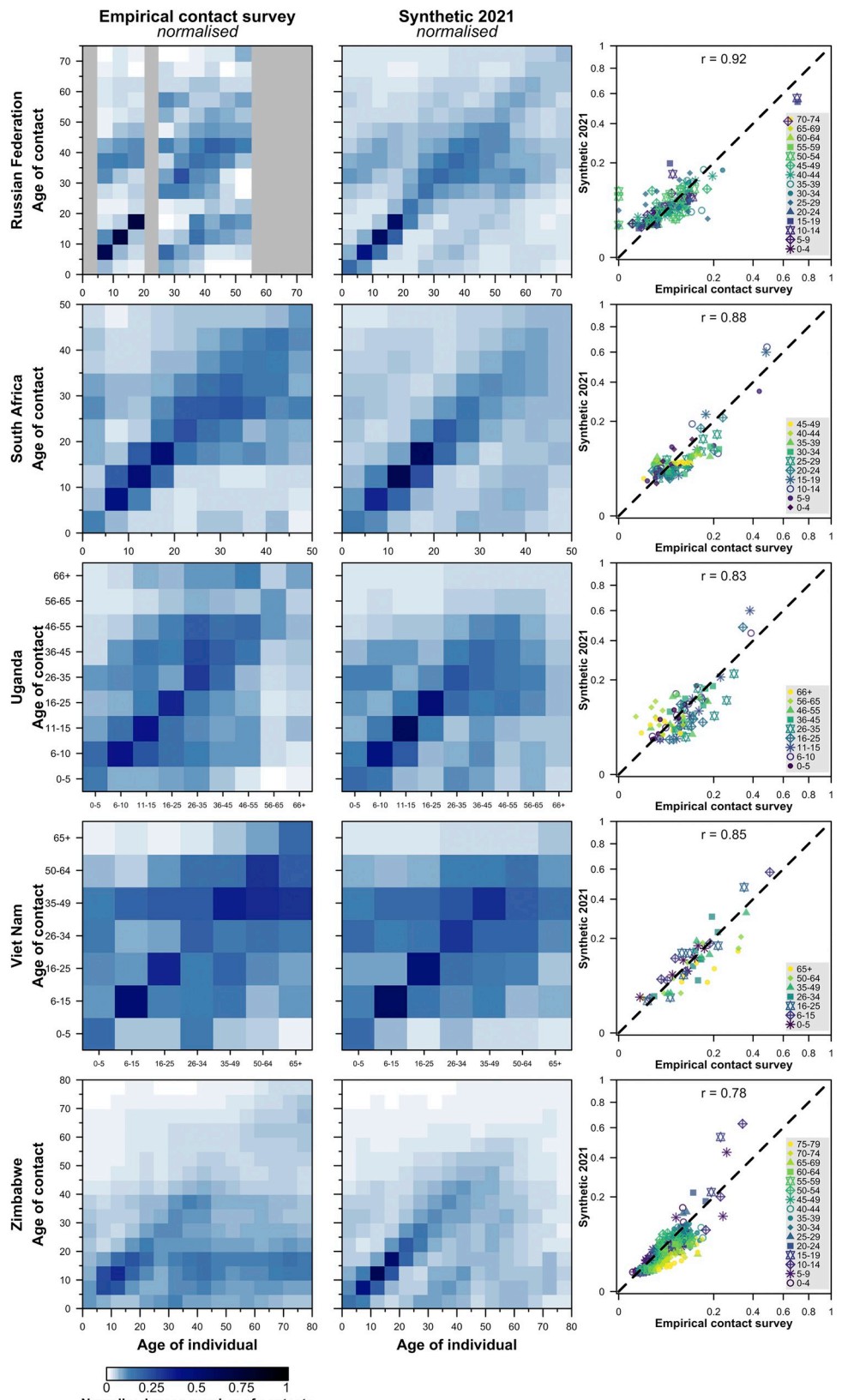

**Fig 3. Comparison of the normalised empirical and synthetic age-specific contact matrices in five geographical regions.** The empirical matrices collected from contact surveys, modelled synthetic contact matrices, and the scatter plots of the entries in the observed (x-axis) and modelled (y-axis) contact matrices are presented. The correlation between the empirical and synthetic matrices are shown. The matrices are normalised such that its dominant eigenvalue is 1. To match the population surveyed in the empirical studies, the contact matrices from rural settings of South Africa, Uganda, Vietnam, and Zimbabwe are presented; and the contact matrices from urban settings of the Russian Federation are presented. No data are available in the grey regions.

(empirical and synthetic), we observed small differences in the modelled reduction in number of cases across the three physical distancing interventions for the COVID-19 pandemic.

The synthetic matrices provide consistency for inter-country comparisons since they are based on common datasets. This is challenging to achieve through empirical data collection (see **Table 2**). For such studies, surveying across the whole population poses several challenges. Establishing a sampling frame and obtaining a sample representative of an entire country's population is expensive and in some regions logistically challenging, so researchers often restrict studies to a particular subpopulation. For instance, many recent empirical contact studies only represent certain subregions of countries rather than entire countries. Sometimes surveys rely on nonprobability sampling techniques [43–45,47,48], e.g., convenience and quota sampling, when probability sampling techniques are not feasible. Paper or online self-reported contact diaries are largely used in social contact surveys. Compared to less common face-to-face interviews, respondent-filled contact diaries have a less demanding data collection procedure but may report a lower response rate [21,58]. Zhang et al. [43] found significantly higher contacts documented by telephone interview than by self-reporting in Shanghai, China. In addition, contact diaries can be administered prospectively or retrospectively (**Table 2**). In Hong Kong, prospective surveys have been shown to be less prone to recall bias compared to their retrospective counterpart [44], but it is often more challenging to find willing participants for prospective surveys. However, a study in Belgium [59] found no appreciable effect between retrospective and prospective surveying. Other methods, e.g., proximity sensors and phone-based GPS trackers or Bluetooth scanners, have also been employed to measure mixing patterns between individuals [60–63] and are forming part of many countries' contact tracing efforts during the COVID-19 pandemic [64], though most have been implemented to protect users' privacy by storing data with the user rather than centrally. When we compared our synthetic matrices with empirical contact matrices from 11 studies using contact diaries, we found broad consistencies between findings from the two approaches. However, there were also differences which might reflect the heterogeneity in methods used to collect empirical data.

Another consideration affecting both synthetic and empirical matrices is that they change over time. Estimating synthetic matrices relies on the POLYMOD contact survey administered more than a decade ago. Another larger contact survey, BBC Pandemic [60,65] conducted in the UK used mobile phone-based GPS tracking instead of diary-based surveys, reported a decrease in contacts among adolescents compared to POLYMOD, which may reflect substitution of face-to-face contacts with electronic communication in this age group. Moreover, in addition to the rural-urban environment, age- and location-specific contact patterns could vary by socioeconomic conditions within countries. More differences are expected as countries implement physical distancing measures to mitigate the COVID-19 pandemic. The COVID-19 pandemic has affected contact patterns, whether through non-pharmaceutical interventions or reactive behavioural changes, in particular, how we come into contact with one another. Baseline, expected contact rates, as those inferred here, are critical for determining the amount of change in contact rates in response to the pandemic. Understanding the impact of the COVID-19 pandemic on contact patterns requires a detailed analysis of contact surveys

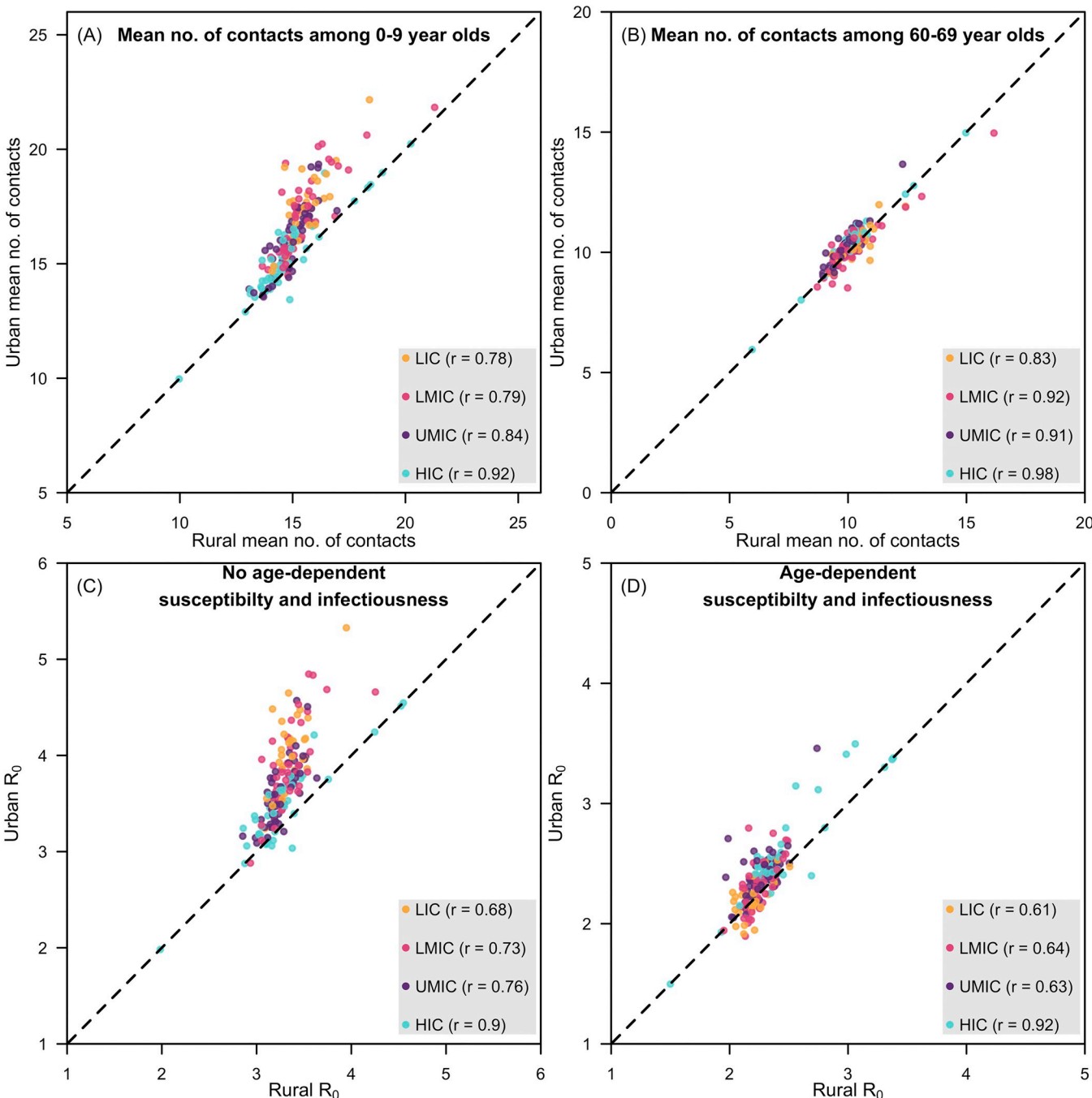

**Fig 4. Mean number of contacts and basic reproduction number between rural and urban settings.** Panels a and b present the scatter plots of the mean number of contacts in younger and older individuals, respectively, in rural (x-axis) and urban (y-axis) settings of a country. Panels c and d present the scatter plots of the basic reproduction number in rural (x-axis) and urban (y-axis) settings of a country without and with age-dependent susceptibility and infectiousness. Geographical regions are grouped as low-income countries (LIC), lower-middle-income countries (LMIC), upper-middle-income countries (UMIC), and high-income countries (HIC), as designated by the World Bank in 2019. Within income group correlations of rural and urban values are presented in the accompanying parentheses.

conducted during the pandemic, taking into account baseline contacts and non-pharmaceutical interventions intensity. Future studies are also needed to quantify the possible long-term behavioural changes.

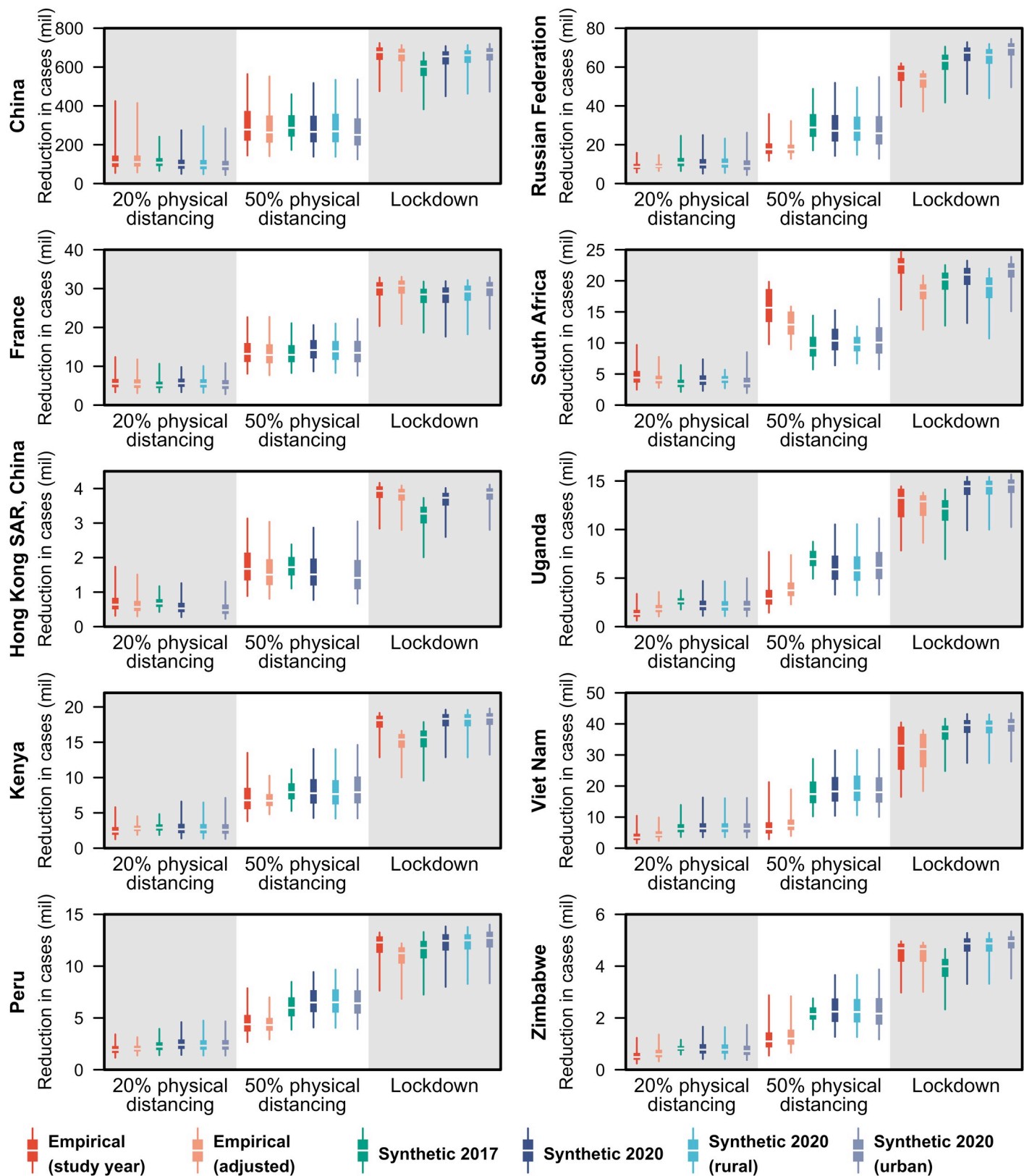

**Fig 5. Reduction in cases due to interventions in models of COVID-19 epidemics under three intervention scenarios in ten geographical regions using the empirical and synthetic matrices.** The reduction in cases in each of the three intervention scenario—20% physical distancing, 50% physical distancing, and lockdown

—against the unmitigated epidemic under different contact matrices is shown in the boxplots with boxes bounded by the interquartile range (25th and 75th percentiles), median in white and, whiskers spanning the 2.5–97.5th percentiles. Six contact matrices were considered in the COVID-19 modelling: the empirically-constructed contact matrices at the study-year and adjusted for the 2020 population, the 2017 synthetic matrices, and the updated synthetic matrices at the national, rural, or urban settings.

Both synthetic and empirical matrices have complementary strengths and limitations. Empirical contact patterns are dependent on the study design and study population, and when the survey is administered. The synthetic contact matrices are constructed using proxies of contacts such as population and household age structures and country characteristics. However, the datasets used to develop these proxy measures (notably population age structure and DHS data) are generally much larger and more nationally representative than most empirical contact studies. To assess the robustness or consistency of the results under different mixing patterns, modellers should consider using multiple contact matrices constructed using different methods for sensitivity analyses.

## Conclusion

In this study, we provide synthetic contact matrices for 177 geographical regions by updating our previous matrices with larger and more recent datasets on population age structure, household, school and workplace composition. The synthetic contact matrices reproduce the main features of the contact patterns in the out-of-sample empirically collected contact matrices.

## Supporting information

**S1 Text. Projecting contact matrices in 177 geographical regions: an update and comparison with empirical data for the COVID-19 era: Supplementary Material: Methods.**
(DOCX)

**S2 Text. Projecting contact matrices in 177 geographical regions: an update and comparison with empirical data for the COVID-19 era: Supplementary Material: results of the household age matrices.**
(DOCX)

**S3 Text. Projecting contact matrices in 177 geographical regions: an update and comparison with empirical data for the COVID-19 era: Supplementary Material: results of the country-specific age- and location-specific contact matrices.**
(DOCX)

**S4 Text. Projecting contact matrices in 177 geographical regions: an update and comparison with empirical data for the COVID-19 era: Supplementary Material results of the comparisons with other contact matrices.**
(DOCX)

## Acknowledgments

We thank Patrick Walker, Oliver Watson and Azra Ghani from Imperial College London, as well as other modellers from around the world, who provided feedback on the 2017 synthetic contact matrices.

The following authors were part of the Centre for Mathematical Modelling of Infectious Disease COVID-19 working group. Each contributed in processing, cleaning and interpretation of data, interpreted findings, contributed to the manuscript, and approved the work for publication: Christopher I Jarvis, Quentin J Leclerc, Jon C Emery, Gwenan M Knight, Amy

Gimma, Simon R Procter, Kathleen O'Reilly, Sophie R Meakin, Charlie Diamond, Stefan Flasche, Billy J Quilty, Anna M Foss, Thibaut Jombart, Katherine E. Atkins, Georgia R Gore-Langton, Adam J Kucharski, James W Rudge, Matthew Quaife, Arminder K Deol, Carl A B Pearson, C Julian Villabona-Arenas, Graham Medley, Alicia Rosello, Hamish P Gibbs, Samuel Clifford, Rein M G J Houben, David Simons, James D Munday, Megan Auzenbergs, Rachel Lowe, Joel Hellewell, Sam Abbott, Damien C Tully, Stéphane Hué, W John Edmunds, Yang Liu, Fiona Yueqian Sun, Oliver Brady, Sebastian Funk, Nikos I Bosse, Akira Endo, Timothy W Russell, Emily S Nightingale.

## Author Contributions

**Conceptualization:** Kiesha Prem, Alex R. Cook, Mark Jit.

**Data curation:** Kiesha Prem, Kevin van Zandvoort.

**Formal analysis:** Kiesha Prem, Kevin van Zandvoort.

**Funding acquisition:** Mark Jit.

**Investigation:** Kiesha Prem, Kevin van Zandvoort, Petra Klepac, Rosalind M. Eggo, Nicholas G. Davies, Alex R. Cook, Mark Jit.

**Methodology:** Kiesha Prem, Kevin van Zandvoort, Petra Klepac, Rosalind M. Eggo, Nicholas G. Davies, Alex R. Cook, Mark Jit.

**Project administration:** Mark Jit.

**Resources:** Mark Jit.

**Software:** Kiesha Prem, Petra Klepac, Rosalind M. Eggo, Nicholas G. Davies.

**Supervision:** Petra Klepac, Alex R. Cook, Mark Jit.

**Validation:** Kiesha Prem, Kevin van Zandvoort.

**Visualization:** Kiesha Prem, Kevin van Zandvoort.

**Writing – original draft:** Kiesha Prem, Alex R. Cook, Mark Jit.

**Writing – review & editing:** Kiesha Prem, Kevin van Zandvoort, Petra Klepac, Rosalind M. Eggo, Nicholas G. Davies, Alex R. Cook, Mark Jit.

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
