## [Decision Letter · Decision Letter 0]

3 Mar 2021

Dear Dr. Jit,

Thank you very much for submitting your manuscript "Projecting contact matrices in 177 geographical regions: an update and comparison with empirical data for the COVID-19 era" for consideration at PLOS Computational Biology. As with all papers reviewed by the journal, your manuscript was reviewed by members of the editorial board and by several independent reviewers. The reviewers appreciated the attention to an important topic. Based on the reviews, we are likely to accept this manuscript for publication, providing that you modify the manuscript according to the review recommendations.

With apologies to the authors for the long delay in this review. We have received three reviews that are complimentary of the work and I believe the concerns can be addressed with minor revisions. Specifically, both R3 and R4 make several minor comments that should be addressed in a revision.

R2 has suggested that some additional detail be provided on the out-of-sample fits. This could be done both quantitatively as the reviewer suggests, or qualitatively - e.g. describing the fit in more detail. For example, Figure 2A is carrying a lot of the weight for the claim the the out-of-sample comparison is good. It seems that there is some additional nuance that could be discussed here and additional information that could be conveyed in the right-hand column of 2A. For example -- consider using different plotting characters (solid circles, open triangles, open squares) to represent matrix cells that are on, above, and below the diagonal, respectively. This would help to visualize whether or not the correlation is dominated by fits to different components of the contact matrix (for example the Shanghai matrix looks like it is dominated by good fit along the diagonal for young ages, but the off diagonals are mismatched - empirical is stronger above the diagonal, synthetic stronger below -- and poorly correlated for older ages; while other matrices appear to be very well correlated). This is just an example of a change to the plot, but the reviewers have requested some more substantive discussion of the out-of-sample fit, which I think would strengthen the manuscript.

Sincerely,

Matthew (Matt) Ferrari

Associate Editor

PLOS Computational Biology

Virginia Pitzer

Deputy Editor-in-Chief

PLOS Computational Biology

[LINK]

With apologies to the authors for the long delay in this review. We have received three reviews that are complimentary of the work and I believe the concerns can be addressed with minor revisions. Specifically, both R3 and R4 make several minor comments that should be addressed in a revision.

R2 has suggested that some additional detail be provided on the out-of-sample fits. This could be done both quantitatively as the reviewer suggests, or qualitatively - e.g. describing the fit in more detail. For example, Figure 2A is carrying a lot of the weight for the claim the the out-of-sample comparison is good. It seems that there is some additional nuance that could be discussed here and additional information that could be conveyed in the right-hand column of 2A. For example -- consider using different plotting characters (solid circles, open triangles, open squares) to represent matrix cells that are on, above, and below the diagonal, respectively. This would help to visualize whether or not the correlation is dominated by fits to different components of the contact matrix (for example the Shanghai matrix looks like it is dominated by good fit along the diagonal for young ages, but the off diagonals are mismatched - empirical is stronger above the diagonal, synthetic stronger below -- and poorly correlated for older ages; while other matrices appear to be very well correlated). This is just an example of a change to the plot, but the reviewers have requested some more substantive discussion of the out-of-sample fit, which I think would strengthen the manuscript.

Reviewer's Responses to Questions

**Comments to the Authors:**

Reviewer #1: Reproducibility report has been uploaded as an attachment.

Reviewer #2: This study presents an update to the study by Prem et al., which is a useful tool for projecting contact matrices to countries that don’t have detailed data on contact structures. The original study was published in PLOS Comp Biol. This study makes some incremental but useful advances over the original, including providing breakdowns by urban/rural populations. Overall, this provides a great resource for the modeling community and is something that will continue to get a lot of use. The demonstration that the synthetic matrices yield similar predictions to the empirical matrices in most situations is also a useful validation of the approach, as is the discussion and table exploring possible reasons for discrepancies..

Comments:

1)There are a number of ways in which data could be stratified sub-nationally. Urban/rural makes some sense and lines up with certain available data sources. But even within ‘urban’ areas there could be drastic differences in contact patterns, thinking of more and less affluent areas of major cities like London or New York. I' not sure that much can be done about this, but the authors could discuss additional ways inwhich subnational projections could be generated.

2)The github site contains the synthetic contact matrices for each county and for contact locations in each country. I didn’t see the associated uncertainty estimates, which might be useful to provide for use in certain modeling applications.

Reviewer #3: This manuscript presents updated results from a 2017 work from the same first author. The main changes with respect to the 2017 work include updating the datasets used to 2020 data, the inclusion of additional countries that account for an additional 1.3% of the world population and some methodological improvements in the estimation of the location specific contact matrices.

The methodology is sound and an adequate level of details is provided in order to assess the results, without the need to resort to the original 2017 paper.

I have one major point that I think should be explored and discussed, more specifically:

- When simulating the impact of physical distancing on COVID-19 spread, the authors use percentages of physical distancing are assumed uniform across all locations. However this is usually not the case when intervention strategies are put into place, for example when school closure, teleworking and the closure of leisure activities are implemented. Since contacts at home, school, work and other location for the synthetic matrices are generated according to different methods and assumptions, it would be interesting to understand how reduction in contacts that are location specific affect the results of the simulations. This would help in assess the possible bias that the synthetic matrices induce, especially when are used to assess the impact of intervention strategies.

I also have some minor comments that could be addressed by the authors to improve the manuscript readability that I list below.

- “Assessing such policies requires differences between contact patterns in rural and urban environments to be quantified, which has previously been done only for a few countries (28,29)”. Also Melegaro et al. (already cited as citation 50 in the manuscript) records different contact patterns in rural and urban setting, and could be added to the sentence.

-”In these studies, individuals in rural settings documented more contacts at home than their

urban counterparts”. Why is the number of contacts at home discussed and not the results in terms of total number of contacts? In case there is no specific reason (e.g. relation to “stay at home” interventions), the overall result should be stressed here too.

- The authors link the contacts at work with the matrix of the working population. However, when a participant makes a contact “at work” it could be a contact with a “customer” , who would not report that contact as happening at work. My point is that imposing the structure of the working population both at the participant level and at the contact level may induce a bias in the contact matrix. Could the authors discuss this issue more in depth?

- “We compare the difference in relative reduction of COVID-19 cases between using

the empirical and synthetic matrices in models of COVID-19 epidemics […] using an age-stratified

compartmental model (13,25)”. The authors should specify here that the model is presented in more depth in the additional material.

- “We model an unmitigated epidemic and three intervention scenarios: 20% physical distancing, 50% physical distancing, and shielding.” Are these assumed reduction in physical distancing in line with the reduction measured from empirical contact data measured during and after lockdown? There are several examples in literature that could be used as a reference ( Zhang et al. 2020, Jarvis et al. 2020, Coletti et al. 2020, Latsuzbaia et al. 2020).

-”Prospective surveys have been shown to be less prone to recall bias compared to their retrospective counterpart (43), but it is often more challenging to find willing participants for

prospective surveys.” However, a study by Beutels et al. 2006 found no appreciable effect of retrospective vs prospective surveying, so this issue may be different, depending on the survey setting.

References

Beutels P.et al. Social mixing patterns for transmission models of close contact infections: exploring self-evaluation and diary-based data collection through a web-based interface. Epidemiology & Infection.

Zhang, J. et al. Changes in contact patterns shape the dynamics of the covid-19 outbreak in china. Science

Jarvis, C. I. et al. Quantifying the impact of physical distance measures on the transmission of COVID-19 in the UK. BMC Medicine

Coletti P. et al. CoMix: comparing mixing patterns in the Belgian population during and after lockdown. Scientific Reports.

Latsuzbaia, A., Herold, M., Bertemes, J.-P. & Mossong, J. Evolving social contact patterns during the covid-19 crisis in Luxembourg. PLOS ONE 15,

Reviewer #4: This is an impressive update to the earlier version of the matrices.

Detailed comments.

Abstract: “We found that the synthetic contact matrices reproduce the main traits of the contact patterns in the empirically-constructed contact matrices.” Needs to be more specific - what is the measure of similarity and how good is it.

No page numbers or line numbers! Using page numbers from the pdf.

Page 9, Line +3; These contact matrices have been useful and have been used to explore policy space. However, it is not clear that their shape is preserved in the presence of a severe pathogen. The coix study from the UK gets to this to some degree. The work presented here is still incredibly useful, but this section does need some caveats so prevent modelling studies making overly strong assumptions.

10,+10; use of “proclivity” not clear here

10,+13; How is the “spatial” smoothness. That doesn’t quite make sense to me.

13,+14; Suggest a zenodo archive of the github with a doi as a permanent record of the code that goes with the paper

13,-6; shielding: this needs a little more discussion. There isn’t really evidence for this as an effective policy. It represents a desirable outcome, but wasn’t really achieved by a deliberate policy anywhere. School closure as a reduction in children’s contacts would be a better generic example.

15,+10; is HAM defined? What was the range of correlations? This is the key bit of the paper. I suggest a supp in figure of actual versus predicted heat-map-grid charts of predicted versus observed for the polymod data

15,-11; please state the magnitude of the difference, perhaps in % peak of epidemics or % change in time to peak, or in % change in attack rate. As with the original paper, this is an important resource, but its also a resource we will want to improve on over time. It’s therefore important to identify numerical targets for those improvements.

17,-2; The discussion paragraph on time is important and should also reflect transient changes during, especially during the key initial phase of a pandemic.

**Have all data underlying the figures and results presented in the manuscript been provided?**

Reviewer #1: Yes

Reviewer #2: Yes

Reviewer #3: Yes

Reviewer #4: Yes

PLOS authors have the option to publish the peer review history of their article (what does this mean?). If published, this will include your full peer review and any attached files.

Reviewer #1: **Yes: **Anand K. Rampadarath

Reviewer #2: No

Reviewer #3: No

Reviewer #4: No

Figure Files:

Data Requirements:

Reproducibility:

References:

---

## [Editor Report · Decision Letter 1]

3 May 2021

Dear Dr. Jit,

Thank you very much for submitting your manuscript "Projecting contact matrices in 177 geographical regions: an update and comparison with empirical data for the COVID-19 era" for consideration at PLOS Computational Biology. As with all papers reviewed by the journal, your manuscript was reviewed by members of the editorial board and by several independent reviewers. The reviewers appreciated the attention to an important topic. Based on the reviews, we are likely to accept this manuscript for publication, providing that you modify the manuscript according to the review recommendations.

I thank the authors for their careful consideration of the reviewers' comments. I am satisfied with the work the authors have done to address these concerns, but I am concerned that a stronger case must be made in the main text about the qualitative and quantitative assessment of fit between the empirical and modeled matrices (presented in Supplementary Tables 4 and 5). These tables are quite helpful, but also put the magnitude of the fit into context (sometimes the fit is poor e.g. Zimbabwe in Table 5). At present there are only quite general references to the supplement, rather than explicit statements directing the reader to these specific results. I would encourage the authors to add specific text in the results to direct the reader to both Tables 4 and 5 (rather than just the corresponding sections) including a summary statement of what is to be found in the tables; e.g. a qualitative assessment of the fit and characteristics of each study (as in L348-354) and a quantitative comparison of the symmetry as a summary measure of fit.

Sincerely,

Matthew (Matt) Ferrari

Associate Editor

PLOS Computational Biology

Virginia Pitzer

Deputy Editor-in-Chief

PLOS Computational Biology

[LINK]

I thank the authors for their careful consideration of the reviewers' comments. I am satisfied with the work the authors have done to address these concerns, but I am concerned that a stronger case must be made in the main text about the qualitative and quantitative assessment of fit between the empirical and modeled matrices (presented in Supplementary Tables 4 and 5). These tables are quite helpful, but also put the magnitude of the fit into context (sometimes the fit is poor e.g. Zimbabwe in Table 5). At present there are only quite general references to the supplement, rather than explicit statements directing the reader to these specific results. I would encourage the authors to add specific text in the results to direct the reader to both Tables 4 and 5 (rather than just the corresponding sections) including a summary statement of what is to be found in the tables; e.g. a qualitative assessment of the fit and characteristics of each study (as in L348-354) and a quantitative comparison of the symmetry as a summary measure of fit.

Figure Files:

Data Requirements:

Reproducibility:

References:

---

## [Editor Report · Decision Letter 2]

20 May 2021

Dear Dr. Jit,

We are pleased to inform you that your manuscript 'Projecting contact matrices in 177 geographical regions: an update and comparison with empirical data for the COVID-19 era' has been provisionally accepted for publication in PLOS Computational Biology.

Best regards,

Matthew (Matt) Ferrari

Associate Editor

PLOS Computational Biology

Virginia Pitzer

Deputy Editor-in-Chief

PLOS Computational Biology

I thank the authors for their revisions and am happy to recommend this manuscript for acceptance.

---

## [Editor Report · Acceptance letter]

20 Jul 2021

PCOMPBIOL-D-20-01307R2 

Projecting contact matrices in 177 geographical regions: an update and comparison with empirical data for the COVID-19 era

Dear Dr Jit,

I am pleased to inform you that your manuscript has been formally accepted for publication in PLOS Computational Biology. Your manuscript is now with our production department and you will be notified of the publication date in due course.

With kind regards,

Olena Szabo
